# The Relationship between Bachelor’s-Level Nursing Roles and Job Satisfaction in Nursing Homes: A Descriptive Study

**DOI:** 10.3390/ijerph21020238

**Published:** 2024-02-18

**Authors:** Marijke Mansier-Kelderman, Marleen Lovink, Anke Persoon

**Affiliations:** 1School of Health Studies, HAN University of Applied Sciences, 6503 GL Nijmegen, The Netherlands; marijke.mansierkelderman@han.nl; 2Department of Primary and Community Care, Research Institute for Medical Innovation, Radboud University Medical Center, University Knowledge Network for Older Adult Care Nijmegen (UKON), 6500 HB Nijmegen, The Netherlands; marleen.lovink@radboudumc.nl

**Keywords:** nurses, nursing homes, nursing roles, job satisfaction

## Abstract

The greatest shortages in the nursing discipline are expected in nursing homes. Although job satisfaction is an important factor in the retention of Bachelor’s-level nurses (BNs), little is known about the relationship between the BN roles that are performed on a daily basis and job satisfaction. A cross-sectional, descriptive, questionnaire study was conducted which was based on a convenience sample. The extent of performing seven BN roles was assessed by a self-developed questionnaire. Satisfaction was investigated at three levels: satisfaction with the BN role performance, satisfaction with the work packet (the combination of all roles performed) and satisfaction with job function (all things considered). Respondents (N = 78) were satisfied with the performance of all BN roles (range 3.71–4.42), generally satisfied with the work packet (M = 3.96; SD = 0.96) and neutral about the job function (M = 3.15; SD = 1.12). Not one single BN role correlated with job satisfaction, and the work packet (as a combination of all roles) was significantly related to job satisfaction (*r* = 0.551; *p* = 0.000). Four BN roles correlated significantly with satisfaction with the work packet, of which one was meaningful, the role of reflective Evidence-Based Practice professional (*r* = 0.476; *p* = 0.000), and three roles related less strongly: the roles of Organiser (*r* = 0.364; *p* = 0.001), Communicator (*r* = 0.224; *p* = 0.049), and Professional and Quality Enhancer (*r* = 0.261; *p* = 0.021). It is important for nurses to create interesting packets of BN roles for themselves. For nurses and care managers, it is essential to create interesting BN descriptions, with highly recognisable BN roles in the work packet, and to stimulate a work environment in order to enhance job satisfaction.

## 1. Introduction

According to estimates from the WHO, there will be a shortage of 5.7 million nurses and midwives by 2030 [1]. Although there is a shortage of staff throughout the entire healthcare sector, the greatest shortage of nursing professionals and formal caretakers is expected within the context of nursing homes and home healthcare [1]. Many strategies are known to help prevent nursing shortages, including staff retention (e.g., by offering career-development opportunities) [2,3], the provision of social support by colleagues and supervisors [4], the creation of a challenging work environment, the reduction of emotional distress, the prevention of disappointment about the day-to-day reality of nursing and the limitation of a culture of hierarchy and discrimination [5]. One overarching theme in the retention of nurses is job satisfaction, a complex concept which can be defined as nurses’ positive-feeling response to the work conditions that meet his or her desired needs as the result of their evaluation of their work experience [6]. Studies have demonstrated that greater job satisfaction reduces attrition and turnover in the healthcare sector, in addition to working as an attractive factor in the intention to stay [7]. It is thus important to encourage job satisfaction [3,8,9]. Job satisfaction has been studied widely and is a multifaceted concept and influenced by a variety of extrinsic factors (e.g., salary and rewards), and may be even more strongly influenced by intrinsic factors (e.g., professional identity and awareness about improving job satisfaction) [10], as well as by the fulfilment of the needs and wishes of patients within the work setting, happiness with regard to working conditions, and job value [6]. 

In the nursing-home setting, the nursing discipline is represented primarily by certified nurse assistants [11] (European Qualification Framework/EQF Level 3), with some vocationally trained registered nurses (EQF Level 4) and, recently, a number of Bachelor’s-level nurses (BNs) (EQF Level 6) [12], who are able to supply valid coaching for certified nursing assistants [13]. In this study, we focus on BNs working in nursing homes, especially on the importance of retaining them in this setting. Aloisio et al. have conducted a systematic review on nurses’ job satisfaction specifically in long-term care and identified seven individual factors: age, health status, self-determination/autonomy, psychological empowerment, job involvement, work exhaustion, and work stress [14]. One related factor to the psychological empowerment and job involvement of the nurse is professional identity, which is defined as a set of characteristics and behaviours that BNs learn during their academic training and continue to develop in the field [10,15,16]. Professional identity is fundamental to nursing practice, sets basis for the professionalisation of nursing and is defined as the attitudes, values, knowledge, beliefs and skills that are shared with others within a professional group in the work place and, most importantly, is a factor of influence in job satisfaction [16]. However, baccalaureate-educated nurses working in nursing-home settings is a rather new phenomenon and professional identity is just not fully grown, and literature about their identity, tasks and competencies is scarce. As the BN is an added extra to a nursing-care team of lower-educated staff members, collaboration between each other and differentiation in roles is still on going and tasks are often allocated inappropriately [17]. Backhaus et al. conclude in an expert consensus study that a “one size fits all” approach to employing BNs is not possible [11]. We recognise in our regular contact with BNs that the results from two older studies still stand: that baccalaureate-prepared RNs in nursing homes have a less strongly differentiated role relative to those in hospitals [18] and that there is a lack of appropriate role models in nursing-home staff [19]. Still, managers who were working with BNs successfully articulated the importance of BNs in nursing homes: ‘Nursing homes will need to address a rising demand for services, more complex resident needs and an increase in patient flow in the near future. Additionally, working in nursing homes is becoming increasingly complex due to a focus on resident-centred care, technological innovations and the expectation for staff to partner with residents and their families throughout the care delivery process’ [11]. To perform a BN role, competencies are required, which are defined as ‘the proven ability to use knowledge, skills and personal, social and/or methodological abilities, in work or study situation and in professional and personal development’ [20]. 

Given that job satisfaction is essential to the retention of BNs in nursing homes, and given that professional identity influences job satisfaction, an in-depth study of the relationship between the job satisfaction of BNs and the fulfilment and performance of BN roles would be highly valuable. To our knowledge, no previous studies have examined this issue, despite its importance in light of the major staffing shortages expected, particularly within the context of nursing homes. The aim of this study is to gain insight into the relationship between the extent to which the various BN roles are performed in daily practice and job satisfaction amongst BNs in nursing homes. 

## 2. Materials and Methods

### 2.1. Setting and Sample

A cross-sectional, descriptive study was performed to examine the relationship between the performance of BN roles and job satisfaction. In the study, BNs completed a one-time quantitative questionnaire. Data were collected from March 2022 to the end of April 2022. The sample was drawn from the general population of BNs working in nursing homes in the Netherlands, which was estimated at a total of 2000 BNs in 2021 [21]. To be included in the study, participants were required to speak Dutch and have at least one year of experience in the nursing-home setting. No restrictions were imposed with regard to position or job function. Convenience sampling was used to obtain as many eligible participants as possible. An invitation flyer was sent to BNs through several nursing homes affiliated with six academic elder-care networks in the Netherlands (SANO) as well as through social media (LinkedIn and Facebook). Participants were able to access the questionnaire through a link and a QR code on the flyer. Due to the mode of distribution (flyers and social media), it is not possible to determine response rate: it is unknown how many people have read the invitation flyer or the message on social media.

### 2.2. Questionnaire

The questionnaire addressed two parameters: the extent to which BNs performed the various roles, and job satisfaction. 

The former parameter was measured by examining the roles as described in the Dutch BN education programme [22]. The Dutch association of nursing educators (LOOV) developed a national education programme to prepare students for their BN roles [22] based on the Dutch national professional BN profile, developed by an expert panel in 2012 [23]. Based on this profile, LOOV developed the education programme written in seven BN roles and 23 required competencies, the structure and jargon being in line with internationally terminology of the CanMEDS’ roles of doctors [24]. Although the roles of doctors and nurses are mostly alike, for example the roles of Communicator, Collaborator and Health Advocate, the interpretation and content of the roles differ, as do their required competencies. The BN roles encompass (1) Healthcare Provider, (2) Communicator, (3) Collaborator, (4) Reflective EBP Professional, (5) Health Advocate, (6) Organiser, and (7) Professional and Quality Enhancer; see Box 1 [22]. In the education programme, each role is described by two to four competencies; for a total of 23 competencies, see Box 1. In the questionnaire, respondents were asked to indicate the amount of time they spent on each competency (‘I spend a lot of time on this competency in my job’). Their responses were scored on a five-point Likert scale, ranging from 1 (strongly disagree) to 5 (strongly agree). 

Box 1Roles and competencies of BNs in the Netherlands (2018) [22].

**BN Role**

**Competency**

**Description**
Healthcare ProviderClinical ReasoningThe continuous process of collecting and analysing data for the purpose of determining the questions and problems of the care user and deciding upon appropriate care results and interventions.
Implementing CareProviding integrated care by independently performing all nursing procedures (including reserved and risky procedures) that occur in complex care situations with due observance of current legislation and regulations and from a holistic point of view.
Strengthening Self-managementSupporting the self-management of people, their loved ones and their social network, with the aim of enabling them to maintain or improve their daily functioning in relation to health and illness and quality of life.
Indicating CareEstablishing and describing the nature, duration, scope and purpose of the required care and arranging for the care to be provided, in conjunction with the care user, on the basis of patient problems that have been diagnosed or potential problems that require further examination and diagnosis.CommunicatorIndividual-focused CommunicationActively listening to the care user, asking them for information and helping them make care-related decisions, and treating the care user as a unique person; acting as a natural guide, coach, expert or advisor, depending on the time and circumstances.
Use of ICTUsing the latest information and communication technologies and providing remote care (e-health) to supplement personal contact with the care user.CollaboratorProfessional RelationshipEstablishing and maintaining contact with the care user, their loved ones and their social network, maintaining long-term care relationships and carefully scaling back these relationships where necessary.
Joint Decision makingSystematically engaging in dialogue with the care user and their loved ones regarding the nursing care to be provided and making sure that clear consideration is taken regarding different sources of knowledge and the values held by the care user during the decision-making process.
Multidisciplinary CollaborationApplying one’s own nursing expertise and collaborating on an equal basis with people from one’s own discipline and other disciplines within and beyond the healthcare sector in relation to multidisciplinary care and other care as well as treatment goals.
Continuity of CareSharing knowledge and information with a view to guaranteeing the continuous involvement of the required care providers in providing care to the care user over time.Reflective EBP ProfessionalInvestigative AbilityDemonstrating a critical investigative and reflective attitude in care situations and with care-related and organisational issues, justifying one’s actions based on various knowledge sources, adopting a methodical approach based on thorough problem analysis and completing the research cycle with a view to improving a specific professional situation.
Use of EBPIn conjunction with the care user (and/or their network), colleagues and other disciplines, assessing (1) recent nursing knowledge actively sought in the scientific literature, guidelines or protocols, (2) professional expertise and (3) the personal knowledge, wishes and preferences of the care user and/or their network.
Professional DevelopmentDemonstrating active and critical behaviour in order to improve and maintain one’s nursing expertise and that of others, and making an active contribution to searching for, developing and sharing knowledge and new forms of knowledge.
Professional ReflectionPerforming a critical assessment of one’s own nursing performance in relation to the professional code and values and putting forward carefully considered arguments during monodisciplinary and multidisciplinary discussions on care users, taking into account the emotions and interests of the care user based on the understanding of care as a moral and ethical practice.Health AdvocatePreventive AnalysisAnalysing the care user’s behaviours and environment that lead to health-related problems for the care user and target groups.
Promoting a Healthy LifestyleOffering support in achieving a healthy lifestyle in relation to potential and existing health problems.OrganiserNursing LeadershipTaking the initiative in managing one’s own area of expertise based on an enterprising-, coaching- and results-oriented attitude.
Coordination of CareTaking the initiative in organising care so that it proceeds smoothly according to the care plan in conjunction with the care user and in coordination between the various care providers and care organisations.
Promoting SafetyMaking a continuous and methodical contribution to promoting and ensuring the safety of care users and employees.
Entrepreneurship in NursingConsidering and acting in accordance with financial–economic and organisational interests within the different contexts of care.Professional and Quality EnhancerProviding Quality of CareMonitoring, implementing and safeguarding the quality of nursing care in a methodical and critical manner.
Participating in the Quality ProcessMaking a proactive contribution to quality assurance within the care organisation.
Professional ConductActing and behaving in accordance with the professional standard and professional code, taking responsibility for all one’s actions and demonstrating professional pride.Note. EBP: Evidence-Based Practice


Job satisfaction was investigated at three levels: (a) satisfaction with performing specific competencies (resulting in 23 ratings: ‘Performing this competency contributes to my satisfaction.’); (b) satisfaction with the work packet, the total of all roles performed (single item, at the end of the questionnaire: ‘How satisfied are you with the totality of the work packet in your job?’); and (c) satisfaction with job function (single item: ‘How satisfied are you with your job as a whole, all things considered? (e.g., salary, appreciation, workload, career opportunities)’). Satisfaction with work packet stands for the satisfaction with the content of all nurses’ performances throughout the day, and relates to roles, tasks and responsibilities. The third-level item (satisfaction with job function) was derived from a survey [25]. The second-level item (satisfaction with the work packet) was constructed and developed after feedback from experts in the pilot (see Methods). A five-point Likert scale was used for all questions, ranging from 1 (highly dissatisfied) to 5 (highly satisfied). Satisfaction levels a, b and c ranged from 1 (highly disagree) to 5 (highly agree).

Demographic data (e.g., gender, age, educational degree and years of work experience) were collected to describe the sample. Open-ended questions were used to collect information on position titles and job descriptions, given the wide variety of job functions and job descriptions for BNs in nursing homes. 

### 2.3. Pilot Study

Before the questionnaire was distributed, it was pretested by five BNs and two nurse researchers. Based on their feedback, textual changes were made regarding the difficulty of the language, the comprehensibility of the questions and the length of the questionnaire. They recommended adding a level of satisfaction between satisfaction with role performance and satisfaction with job function, in order to obtain more information about the possible mechanisms underlying job satisfaction.

### 2.4. Reliability

Because the questionnaire was composed partly of items from previously validated questionnaires and partly of items developed by the researchers, the questionnaire was tested for internal consistency. To assess intra-rater reliability, a retest opportunity was provided, with participants being asked to complete the questionnaire a second time after three weeks. By voluntarily providing their e-mail address, respondents indicated their willingness to participate in the retest and their consent for their email addresses to be used only for this purpose. 

### 2.5. Data Analysis

Incomplete questionnaires were not included in the analysis, as the absence of answers to the last two questions made it impossible to examine the relationship between the extent of performing BN roles and satisfaction with the work packet and job function. All analyses were performed using the International Business Machine Statistical Packet for the Social Sciences (IBM SPSS) Statistics, version 25 (Armonk, NY, USA).

Descriptive statistics (means, range, standard deviations and percentages) were used to describe the demographics of the sample (age, gender, years of formal training, years of experience as a BN and position title). The frequency and mode of each competency (performance and contribution to satisfaction) were determined. The extent of BN role performance and satisfaction with performing a role were calculated as the mean scores (and standard deviations) of the amounts of time spent performing the relevant competencies.

The internal consistency of the questionnaire was tested by calculating Cronbach’s alpha, with values between 0.70 and 0.95 interpreted as positive [26]. Intra-rater reliability was calculated as the intraclass correlation coefficient (ICC agreement, two-way), with values between 0.00 and 0.20 interpreted as slight, between 0.20 and 0.40 as fair, between 0.40 and 0.60 as moderate, between 0.61 and 0.80 as substantial, and between 0.81 and 1.00 as almost perfect [27]. 

The correlation between the extent of BN role performance and two levels of satisfaction (work packet and job function) were calculated using Spearman’s Rho. A correlation was perceived as significant if *p* < 0.05 and strongly significant if *p* < 0.01. Correlations higher than 0.4 were considered meaningful [28]. Correlations between satisfaction with the work packet and satisfaction with the job function were also calculated using Spearman’s Rho. 

### 2.6. Ethical Considerations

The research ethics committee of the region Arnhem–Nijmegen, the Netherlands, declared this study not to be subject to the Medical Research Involving Human Subjects Act (registration number: 2022-13481). The study was conducted in conformity with the Declaration of Helsinki [29]. The European General Data Protection Regulation (GDPR) was also followed in this study [30]. The completion of the questionnaire was anonymous, unless respondents indicated that they were willing to participate in the retest component. The email addresses that were saved for the retest were deleted after the second questionnaire. Informed consent statements and all other data were stored in a secure online database at the University Knowledge network for Older adult care Nijmegen (UKON)/Radboud university medical centre (Radboudumc) and will be kept for a maximum of 15 years, in accordance with UKON/Radboudumc guidelines. 

Informed consent was requested in the first item of the questionnaire. If no consent was granted, the questionnaire was discontinued. Participants could stop the questionnaire at any time for any reason without any consequences. 

## 3. Results

### 3.1. Participants and Response Rate

A total of 85 questionnaires were completed. Of these, seven were excluded for not meeting the inclusion criterion of having a Bachelor’s degree in nursing. In all, the questionnaire was opened and only viewed 201 times, filled out up to informed consent 27 times, filled out up to demographic data 18 times, and filled out up to halfway through the questionnaire 16 times. For the reliability test, 19 people completed the questionnaire twice.

### 3.2. Demographic Data

As shown in Table 1, the final sample comprised 78 BNs working in nursing homes. The majority of respondents were women (89.7% women, 10.3% men), and respondents’ ages ranged from 22 to 64 years, with a mean age of 38 years (SD 11.82). Almost half (47.5%) had obtained their Bachelor’s degree within the past five years, and 39.7% had been working as a BN for 2–5 years. Most respondents described their position title as ‘BN’, and most were working as ‘quality nurse’ or as ‘BN with a specialisation’. The respondents included three BNs who were working at a level lower than that for which they had been trained (secondary vocational nurse), and two who were employed in management positions (team coach and location manager).

### 3.3. Reliability

As reported in Table 2, the internal consistency of the questionnaire on the items regarding how much time was spent on the competence and satisfaction with performing the competence was positive (*α* ≥ 0.910). And, internal consistency for the items on satisfaction with the work packet and satisfaction with the job function was doubtful (*α* = 0.697). Of all 78 respondents, 19 completed the questionnaire a second time. As shown in Table 3, the ICC for each role was at least moderate (*r* ≥ 0.4), with the exception of performing the Organiser role, for which the ICC was fair (*r* = 0.304). 

### 3.4. Extent of BN Role Performance

On average, respondents performed all BN roles, although they differed in the extent to which they performed the various roles; see Table 4. The most frequently performed BN role was that of Professional and Quality Enhancer (M = 4.21), and the least frequently performed was that of Health Advocate (M = 3.47). The frequency of performing the several competencies varied considerably, see Table 4. The competency of Nursing Leadership (as an element of the role of Organiser) was the most commonly performed (M = 4.35), and the competency of ICT Use (as an element of the role of Communicator) was performed least commonly (M = 3.09).

### 3.5. Satisfaction in Performing Roles, Work Packet and Job Function 

As described in Table 4, performing the role of Professional and Quality Enhancer contributed the most to satisfaction (M = 4.42), and the role of Healthcare Provider contributed the least (M = 3.71). Satisfaction with performing the several competencies varied more than satisfaction with performing BN roles; see Table 4. More specifically, performing the competency of Nursing Leadership (as an element of the Organiser role) contributed the most to satisfaction (M = 4.62), and the competency of Joint Decision-making (as an element of the Collaborator role) contributed the least (M = 3.00). 

As reported in Table 5, the mean score for satisfaction with the work packet was 3.96 (corresponding to ‘satisfied’). Participants were less satisfied with the job function (all things considered, e.g., salary, appreciation, workload and career) than with the work packet (combination of all roles performed), as rated with a mean score of 3.15 (corresponding to ‘not unsatisfied/not satisfied’); see Table 5.

Four of the seven BN roles were (strongly) significantly correlated with satisfaction with the work packet (*p* < 0.005), although only one had a meaningful strong correlation: the role of Reflective EBP Professional (*r* = 0.476; *p* = 0.000); see Table 6. The other three roles correlating with satisfaction with work packet were the role of Organiser (*r* = 0.364; *p* = 0.001), Communicator (*r* = 0.224; *p* = 0.049), and Professional and Quality Enhancer (*r* = 0.261; *p* = 0.021), as shown in Table 6. 

No significant correlations were found between the extent of BN role performance and satisfaction with the job function; see Table 6. A meaningful strongly statistical correlation was found between satisfaction with the work packet and satisfaction with the job function (*r* = 0.551, *p* = 0.000), see Table 5.

## 4. Discussion

In this study, we examined the relationship between the extent to which nurses working in nursing homes perform BN roles in daily practice and job satisfaction. Job satisfaction was investigated at three levels: satisfaction with performing a BN role, with the work packet (all the performances in daily practice) and the BN job function (all things considered, e.g., salary, appreciation, workload and career). Respondents reported performing all seven BN roles to an extent between average and often in daily practice. They noted that performing the various BN roles contributed to their satisfaction and that they were generally satisfied with their work packets and neutral (‘not unsatisfied/not satisfied’) with regard to their job functions. The extent of performing one role was strongly significant and meaningfully correlated with the work packet: the role of Reflective EBP Professional. The extent of performing three other roles were, although less meaningful, significantly correlated with the work packet: the roles Communicator, Reflective EBP professional, Organiser and Professional and Quality Enhancer. Not one significant correlation was found between the extent of performing any specific BN role and satisfaction with the job function. However, a strongly significant meaningful correlation was found between satisfaction with the work packet and satisfaction with job function.

Looking at the four BN roles which significantly correlated to the satisfaction with the work packet, the role of Reflective EBP Professional stands out as the only role that was significantly correlated and meaningful (*r* > 0.4). In our study, this role is characterized by four competencies, and all four competencies are performed on average and contribute to satisfaction. These results confirm the importance of the trend to position BNs as professionals who reflect, contribute to and encourage evidence-based practice. It is known that a strong culture of EBP enhances job satisfaction for hospital nurses [31]. This competency can be practiced on a patient level, such as through clinical reasoning, but on a team level as well by creating an evidence-based work culture. Enhancing EBP in nursing is important, as several studies suggest that EBP is not integrated into daily practice yet [32,33]. However, Lovink et al. found that developing an evidence-based culture by nurses within care teams of nursing homes is possible, as long as it involves applying creative and motivating methods [34]. Additionally, Handor et al. found five competencies associated with a facilitator role of BNs to promote the development of an effective workplace culture [31]. The role of Professional and Quality Enhancer was reported as the most commonly performed, and the respondents noted that performing this role contributed the most to satisfaction with the work packet, although this relationship was not strong (*r* = 0.261). Enhancing the quality of care can be performed either in direct patient care or as a policy maker; it is unclear in which way participants performed this role and how it contributed to satisfaction with the work packet. The role of Organiser was performed frequently as well, and performing this role contributed to satisfaction as well. Its weak but significant correlation with the work packet might have been due to the low scores assigned to the competency of Entrepreneurship in Nursing, which is not particularly relevant to the Dutch nursing practice in the nursing-home setting. 

Next, our results on the role of Healthcare Provider were somewhat confusing. This was the second least commonly performed role; it contributed the least to satisfaction and it was not related to satisfaction with either the work packet or job function. These results are not in line with those of other studies. Performing patient care should lead to satisfaction and, if BNs are unable to provide the necessary care, this should logically lead to dissatisfaction and even burnout [35,36]. In addition, other studies conducted amongst BNs in the hospital setting have indicated that performing core nursing activities does increase the intention to stay [7]. Our findings may be explained by the fact that we operationalised this role of Healthcare Provider in terms of an array of competencies that includes Indicating Care. This competency received the second-lowest scores of all 23 competencies (only ICT Use scored lower), thus decreasing the average score for the role of Healthcare Provider. In turn, this could be related to the fact that, in the Netherlands, BNs have no role in addressing the care problems of residents and formulating indications for their care. In nursing homes, this task is performed primarily by certified nurse assistants [37]. In light of current increases in multimorbidity and behavioural and neuropsychiatric problems amongst nursing-home residents, however, combined with a focus on facilitating the members of care teams and the introduction of EBP, it is important for BNs to be competent clinical nurses, in both patient care and its indication. The finding that this role contributed the least to satisfaction is thus difficult to understand, as BNs should serve as role models in the provision of care and understand the complexity of care. This should be a prerequisite to assuming tasks such as team coaching, learning, interprofessional collaboration, EBP and family communication. 

All in all, looking at the combination of BN roles performed, it is thus important for nurses to create interesting packets of nursing roles for themselves. For care managers, it is essential to offer interesting BN job descriptions with highly recognisable BN roles in the work packet. 

Finally, the results of this study make it clear that the performance of the work packet as a whole, but not of any individual BN role, is strongly related to job satisfaction. Further, there is a gap between the satisfaction with the work packet, as reported ‘satisfied’, and the ‘neutral’ satisfaction with the job as a whole (not unsatisfied/not satisfied). This finding suggests that factors other than BN roles were reducing the level of job satisfaction, and this could be a potential risk to the retention of BN nurses in the nursing-home setting. Work content and work environment are the strongest determinants of job satisfaction, as shown in the much cited meta-analytic study of Irvine and Evans [38]. Applying these results for our study, we may suggest that participants were satisfied about the work content (roles and performing competencies) but were only neutral about the work environment. It is interesting to focus on the results of the magnet program, a program which is based on research that identified the characteristics of healthcare institutions that succeeded in the recruitment and retention of registered nurses [39]. Nurses working in the environment of magnetism perceived higher job satisfaction and empowerment [40,41,42]. The magnet programme is made up of five core components: transformational leadership; structural empowerment; exemplary professional practice, new knowledge, innovations, and improvement; and empirical quality outcomes [43]. Those components are supported by 14 forces that respond to changing nursing and healthcare environments. In recent years, some studies have found positive patient outcomes in comparison with non-magnet hospitals [44], although studies investigating the effect of patient outcomes do not always provide a clear conclusion [45]. Much is written about the positive impact of magnet programmes on organizational culture, particularly for nurses [46]. Therefore, we recommend creating in nursing homes magnet forces too, to implement and evaluate this highly interesting program; it may help to retain BNs, to improve innovations and to raise the quality of care. This would connect well with the actual movement in the Netherlands in the nursing-home setting which strengthens the leadership and empowerment of certified nurse assistants and (Bachelor-educated) nurses in nursing homes; ‘Landelijke Actieplan Zeggenschap’ [47].

From studies conducted in the hospital setting, we know that years of work experience [48,49], the work environment, team cohesion, working conditions [50,51], control over work and high work demands [7,36,52] are of influence. As reported in a review on satisfaction amongst BNs, factors contributing to job satisfaction differ for BNs working in nursing homes and those employed in acute care settings [53]. For BNs working in nursing homes, a higher age was significantly associated with job satisfaction, although age had no significant effect on job satisfaction for BNs in critical care [53]. The length of work experience has been found to be unrelated to job satisfaction amongst BNs working in nursing homes, whilst being only slightly related to job satisfaction for BNs working in acute-care hospitals [53]. Further in-depth research on the satisfaction of BNs with their job functions could provide more insight into the relationship between nursing activities in daily practice and satisfaction. To this end, a qualitative study design seems most appropriate.

One strength of the study is that this is the first investigation of the relationship between professional BN roles and job satisfaction to be conducted amongst BNs in nursing homes. Another strength is that the questionnaire was tested by and discussed in advance with experts. Moreover, a retest was conducted with satisfying results. It is nevertheless important to reflect on four possible methodological flaws: a small sample size, the representativeness of the sample, the operationalisation of the BN roles and the concept of a work packet. First, the finding that only 2 of 15 correlations were significant could be due to the fact that the study was based on the responses of only 78 participants—the power might have simply been too low. Second, the representativeness of the sample is unclear because the study was based on a convenience sample and the response rate is unknown. It is clear that the sample had a notable over-representation of BNs who had graduated within the past five years. It is not possible to know the direction in which this might have influenced the results. As higher age is significantly associated with higher job satisfaction, it may suggest that older BN are even more satisfied with performing BN roles and the work packet [53]. The over-representation might have been due to the fact that, in the Netherlands, BNs have only recently been introduced to the nursing-home setting, as well as a trend in which many BNs leave the setting after a few years. In addition, the administration of the questionnaire online might have decreased interest in participation amongst older nurses. Moreover, scatter plots of the analyses revealed two clear outliers that had a negative influence on the results. More specifically, two respondents were working as managers, as there were no exclusion criteria based on position, and we invited BNs regardless of their actual functions. Third, we operationalised BN roles in terms of the 23 competencies of BNs, as described in the national BN education profile. Upon closer examination, however, we identified several randomly chosen categorisations. For example, the competency of Nursing Leadership is assigned to the role of Organiser, but it could also have been assigned to other roles as well. This was the case for multiple competencies. Fourth, we studied satisfaction with the work packet according to only one item, and we sought relationships between this item and the seven national BN roles as described by the association of nurse educators. After distributing the questionnaire, we began to wonder whether those roles addressed are characteristic of the work packet of BNs in nursing homes. Other roles might be important, as well, including some not traditionally associated with nursing expertise, e.g., being a team leader, role model and coach within the nursing team [37]. In addition, Nursing Leadership might be experienced as a specific role, and not merely as a competency. For this reason, our questionnaire might have overlooked important roles and competencies that contribute to job satisfaction. A more in-depth study on the roles and competencies performed by BNs in the daily practice could provide insight into what they appreciate in their nursing activities. Such studies could best be conducted according to more qualitative methods, including interviews with and observations of BNs and their colleagues (e.g., certified nurse assistants, physicians, paramedics and managers), as well as nursing-home residents. 

## 5. Conclusions

The BNs in this sample expressed satisfaction with performing all of the specified BN roles, and they were generally satisfied with their work packets. Four BN roles correlated significantly with satisfaction with the work packet, although just one was meaningful: the role of Reflective EBP Professional. The results of this study emphasise that the performance of the work packet as a whole, and not any individual nursing role, is strongly related to job satisfaction. As BNs were just neutral in expressing job satisfaction, it is important for nurses and care managers to create highly recognisable work packets for BN roles for themselves and to enhance a stimulating work environment With regard to the retention of BNs in the nursing-home setting, it is important to identify the factors in-depth that influence satisfaction with job function and the exact relationships between these factors and the extent of BN role performance. This calls for studies of a more qualitative character.

## Figures and Tables

**Table 1 ijerph-21-00238-t001:** Characteristics of respondents (N = 78).

		Frequency (%)	M (SD)
Gender			
	Male	8 (10.3)	
	Female	70 (89.7)	
Age			38.46 (11.8)
	20–29	21 (26.9)	
	30–39	29 (37.2)	
	40–49	7 (9.0)	
	50–59	17 (21.8)	
	60 and older	4 (5.1)	
Years of diploma			2012 (10.0)
	2017–2022	37 (47.5)	
	2012–2016	14 (17.9)	
	2007–2011	11 (14.1)	
	2002–2006	3 (3.8)	
	1997–2001	5 (6.4)	
	1992–1996	3 (3.8)	
	1987–1991	3 (3.8)	
	1982–1986	X	
	1977–1981	2 (2.6)	
Years of working			9.49 (9.9)
	>1	8 (10.3)	
	2–5	31 (39.7)	
	6–10	16 (20.5)	
	11–15	9 (11.5)	
	16–20	3 (3.8)	
	>21	11 (14.1)	
Function name			
	Bachelor’s-level nurse	20 (25.6)	
	Quality nurse	13 (16.7)	
	Bachelor’s degree specialisation	11 (14.1)	
	Senior nurse	9 (11.5)	
	Coordinating nurse	4 (5.1)	
	Management nurse	4 (5.1)	
	Secondary vocational nurse	3 (3.8)	
	Network nurse	2 (2.6)	
	Practice nurse	2 (2.6)	
	Other	10 (12.8)	

**Table 2 ijerph-21-00238-t002:** Internal consistency (N = 78).

	Items	Cronbach’s Alpha
Performances of BN roles ‘I spent a lot of time on this’	23	0.910
Performances of BN roles contributing to satisfaction ‘Performance contributes to satisfaction’	23	0.925
Satisfaction with work packet and job function	2	0.697

**Table 3 ijerph-21-00238-t003:** Intra-rater reliability (N = 19).

	‘I Spend a Lot of Time on This’	‘Performance Contributes to Satisfaction’	Satisfaction
Role:			
- Healthcare Provider	0.701	0.752	
- Communicator	0.628	0.741	
- Collaborator	0.423	0.747	
- Reflective EBP Professional	0.693	0.639	
- Health Advocate	0.475	0.553	
- Organiser	0.304	0.391	
- Professional and Quality Enhancer	0.585	0.581	
Satisfaction:			
- With work packet			0.709
- With job function			0.803

Note. EBP: Evidence-Based Practice.

**Table 4 ijerph-21-00238-t004:** Extent of competency and role performance and contribution to satisfaction (N = 78).

		‘I Spend a Lot of Time on This’ ^a^				‘Performance Contributes to Satisfaction’ ^b^			
BN Role	Competency	M (SD)	Mode	Min	Max	M (SD)	Mode	Min	Max
Healthcare Provider		3.51 (0.90)				3.71 (0.79)			
	Clinical Reasoning	3.77 (1.01)	4.00	1.00	5.00	4.23 (0.82)	4.00	1.00	5.00
	Performing Care	3.55 (1.25)	4.00	1.00	5.00	3.51 (1.17)	4.00	1.00	5.00
	Strengthen Self-management	3.54 (1.15)	4.00	1.00	5.00	3.78 (1.00)	4.00	1.00	5.00
	Indicating Care	3.18 (1.32)	4.00	1.00	5.00	3.31 (1.26)	4.00	1.00	5.00
Communicator		3.62 (0.82)				3.80 (0.77)			
	Individual-focused Communication	4.16 (0.96)	5.00	1.00	5.00	4.15 (0.89)	5.00	1.00	5.00
	Use of ICT	3.09 (1.11)	3.00	1.00	5.00	3.45 (1.03)	3.00	1.00	5.00
Collaborator		3.99 (0.79)				4.13 (0.75)			
	Professional Relationship	3.82 (0.99)	4.00	1.00	5.00	3.91 (0.91)	4.00	1.00	5.00
	Joint Decision Making	3.76 (1.10)	4.00	1.00	5.00	3.00 (0.97)	4.00	1.00	5.00
	Multidisciplinary Collaboration	4.32 (0.89)	5.00	1.00	5.00	4.49 (0.83)	5.00	1.00	5.00
	Continuity of Care	4.05 (1.02)	4.00	1.00	5.00	4.15 (1.05)	5.00	1.00	5.00
Reflective EBP professional		3.76 (0.96)				4.29 (0.85)			
	Investigative Ability	3.67 (1.32)	5.00	1.00	5.00	4.31 (1.01)	5.00	1.00	5.00
	Use of EBP	3.40 (1.20)	4.00	1.00	5.00	4.19 (0.97)	5.00	1.00	5.00
	Professional Development	4.00 (1.08)	5.00	1.00	5.00	4.42 (0.919)	5.00	1.00	5.00
	Professional Reflection	3.96 (1.00)	4.00	1.00	5.00	4.22 (0.921)	5.00	1.00	5.00
Health Advocate		3.47 (1.05)				3.78 (0.98)			
	Preventive Analysis	3.64 (1.18)	4.00	1.00	5.00	3.94 (1.06)	4.00	1.00	5.00
	Promoting a Healthy Lifestyle	3.29 (1.15)	4.00	1.00	5.00	3.62 (1.08)	4.00	1.00	5.00
Organizer		3.96 (0.61)				4.16 (0.50)			
	Nursing Leadership	4.35 (0.82)	5.00	2.00	5.00	4.63 (0.54)	5.00	3.00	5.00
	Coordination of Care	3.92 (1.02)	4.00	1.00	5.00	4.18 (0.88)	4.00	1.00	5.00
	Promoting Safety	4.13 (0.69)	4.00	2.00	5.00	4.28 (0.66)	4.00	2.00	5.00
	Entrepreneurship in Nursing	3.44 (1.08)	4.00	1.00	5.00	3.55 (1.05)	4.00	1.00	5.00
Professional and Quality Enhancer		4.21 (0.68)				4.42 (0.58)			
	Providing Quality of Care	4.31 (0.87)	5.00	1.00	5.00	4.49 (0.79)	5.00	1.00	5.00
	Participating in the Quality Process	4.10 (1.03)	4.00 ^a^	2.00	5.00	4.41 (0.75)	5.00	2.00	5.00
	Professional Conduct	4.23 (0.77)	4.00	1.00	5.00	4.37 (0.65)	4.00	1.00	5.00

Note. EBP: Evidence-Based Practice. ^a^ 1: strongly disagree, 2: disagree, 3: neutral, 4: agree, 5: strongly agree. ^b^ 1: highly unsatisfied, 2: unsatisfied, 3: not unsatisfied/not satisfied, 4: satisfied, 5: highly satisfied.

**Table 5 ijerph-21-00238-t005:** Satisfaction with work packet and satisfaction with job function (N = 78) and their correlation.

	M (SD) ^a^	Mode	Min	Max	*r* (Sig.)
Job Satisfaction					
Satisfaction with work packet	3.96 (0.96)	4	1	5	0.551 (0.000)
Satisfaction with job function	3.15 (1.12)	4	1	5	

^a^ 1: highly unsatisfied, 2: unsatisfied, 3: not unsatisfied/not satisfied, 4: satisfied, 5: highly satisfied.

**Table 6 ijerph-21-00238-t006:** Extent of BN role performance, contribution to satisfaction and correlations between the extent of BN role performance and satisfaction with work packet and satisfaction with job function (N = 78).

	Spending a Lot of Time on BN Role Performance	Satisfaction with BN Role Performance	Satisfaction with Work Packet	Satisfaction with Job Function
BN roles	M (SD) ^a^	M (SD) ^b^	*r* (Sig.)	*r* (Sig.)
Healthcare Provider	3.51 (0.90)	3.71 (0.79)	0.098	−0.083
Communicator	3.62 (0.82)	3.80 (0.770	0.224 *	−0.090
Collaborator	3.99 (0.79)	4.13 (0.75)	0.125	−0.071
Reflective EBP professional	3.76 (0.96)	4.29 (0.85)	0.476 **	0.137
Health Advocate	3.47 (1.05)	3.78 (0.98)	0.056	−0.044
Organiser	3.96 (0.61)	4.16 (0.50)	0.364 **	0.108
Profes. and Quality Enhancer	4.21 (0.68)	4.42 (0.58)	0.261 *	0.160

^a^ 1: strongly disagree, 2: disagree, 3: neutral, 4: agree, 5: strongly agree. ^b^ 1: highly unsatisfied, 2: unsatisfied, 3: not unsatisfied/not satisfied, 4: satisfied, 5: highly satisfied. * indicates significant correlation < 0.05. ** indicates significant correlation < 0.01.

## Data Availability

The data that support the findings of this study are available on request from the corresponding author. The data are not publicly available due to privacy or ethical restrictions.

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
