# Peer review of "The Relationship between Bachelor’s-Level Nursing Roles and Job Satisfaction in Nursing Homes: A Descriptive Study"

_ijerph, 2024, doi:10.3390/ijerph21020238_

Round 1

Reviewer 1 Report

Comments and Suggestions for Authors

Abstract

There is lack of a clear structure stating the goal, methods, results, conclusion. The research results in the abstract are not exactly presented using numerical data.  

Introduction

It is appropriate to analyze the factors affecting job satisfaction in more detail at the beginning. Also to describe in more detail the individual roles of nurses (healthcare provider, communicator, collaborator,...), which are the center of the research, also because the authors of the manuscript do not adress them either in the methodology or in the discussion.

Methods

The description of the tools used is ambiguous. Since some instruments used in the research are constructed by the authors and were not used in other studies, it was appropriate to describe them more precisely. Specify each BN roles – indicate the number of items that make up the role, indicate the Cronbach alpha. Cronbach alpha should have been identified separately for BN roles and contribution to satisfaction – these are separate parameters!!! (tab.2a).

I do not perceive a significant difference between „work packet satisfaction“ and „job function satisfaction“ from the point of view of the questions asked and the presentation and description in section 2.2.

Response rate data (3.1) should be a part of section 2.1 (setting and sample).

Results

Results presented in tables and their description is complicated. It was appropriate to first present all descriptive data for the monitored variables, then present the correlations, and adapt the description accordingly.

The authors state that job satisfaction is analyzed at 3 levels – the first level – „satisfaction with performing specific competencies“ however is not analyzed – it is presented descriptively, correlations with BN roles are absent.

Discussion

„No significant correlations were found between the extent of performing any specific BN role and satisfaction with job function, and only one role (Reflective EBP Professional) was related to satisfaction with work packet“. I do not agree with this summary – in table 3 authors present 4 BN roles that are significant, although with a lower value of the r coefficient. Therefore, the discussion is not presented correctly.

In the discussion the authors also adress areas that they did not evaluate – e.g. the age of the respondents – this parameter is not analyzed in the research.

The research has a large number of limitations - as the authors state in the discussion – but many of them could have been solved before data processing – e.g. elimination of outliers, it was not appropriate to include respondents who have a manager nurses, etc. 

Reviewer 2 Report

Comments and Suggestions for Authors

Dear authors, thanks for the opportunity to review this primary research. Although you have done excellent work, some issues should be addressed.

1.      SCOPE: the manuscript is in line with the thematic scope of the IJERPH.

2.      TITLE: the design of research should be mentioned: “An explanatory study”

3.      ABSTRACT: the length and the quality are correct. Some minor clarifications should be added!

-         Please, provide some significant associations (related to the title and conclusion) showing the level of significance (mean, SD, min -max & p=values) making it more interesting for international readers.

-         The conclusion should reflect the entire findings of the study, in line with the title and the results and less information about the future implications.  

-         Not acronyms in the abstract section (e.g. EBP)

KEYWORDS: The number of keywords is acceptable.

4.      INTRODUCTION: the introduction section is not clear!

-         Lines 56-58, the authors mention that: “The professional identity of BNs can be derived from seven roles 56 formulated by the Canadian Medical Education Directions for Specialists (CanMEDS), 57 which have received international recognition [16] and have been described for several 58 groups of healthcare professionals.”

-         However, the CanMEDS is a framework established by the Royal College of Physicians and Surgeons of Canada to improve patient care by optimizing physician training. Thus the CanMEDS Framework identifies and describes physician abilities required to successfully meet the health care needs of those they serve, and NOT related to nursing roles and tasks. Homecare nurses at EQF 6 (BN level) are not trained during their studies in these 7 CanMEDS Roles. The standards of home care provision are widely recognized by The International Home Care Nurses Organization (IHCNO) https://www.ihcno.org/resources1

-         Ref 18 is not accessible for the reviewers to examine the education curriculum mentions 23 competencies, classified into the seven CanMEDS roles 23 competencies!

-         In this case that A FUTURE-PROOFPROGRAMME PROFILE 4.0 is offered only in the Netherlands, then authors should provide information on how this MEDICAL program (CanMEDS) is related to A FUTURE-PROOFPROGRAMME PROFILE 4.0 and how adapted for Nurses (validity and reliability and cultural adaprion….) in order to increase nurses competency as a “Nurse Expert”. In this context, the aim of this study should be focused on whether a competent nurse (as an Expert) provides a better quality of homecare in comparison to a lower level of EQF (practitioners, eg.) and most importantly to evaluate this program -A FUTURE-PROOFPROGRAMME PROFILE 4.0 by recruiting patients receiving and not nurses.  

-         In general, the “Introduction” section should justify the hypothesis why is so important for this study to be conducted!

5.      METHOD:  sub-2.3

-         2.3 Datata analyses” should be presented after the Section “Ethical considerations”

-         The evaluation of A FUTURE-PROOFPROGRAMME PROFILE 4.0 (e.g. construct validity of the translated version) of each scale used should be mentioned accordingly!

6.      RESULTS: the results of the focus study are not properly presented and described. The presented data are insufficient to draw conclusions due to several limitations.

-         The authors mention in sub 3.1 (line 185) that “The response rate is unknown”….

-         Sample justification is not provided!

-         Power analysis should be applied before the Pilot since the sample size is known (2,000 BNs in 2021 BNs working in nursing homes in the Netherlands) and to justify whether the sample of 78 participants is enough for 2,000 BNs ….

-         In general, the methodological issue is not the “small sample size” but the “low response rate” as regards the generalization of the conclusions!

-         Also, I think that the main limitation is the statistical analysis, particularly in cases where the variable includes less than 5 items per cell (participants), the comparisons between the two variables extract at least “crude” results according to the SPSS instructions

-         Although the authors mention these limitations in the relevant section, it remains an issue!

-         How many of the 78 participants have a degree in A FUTURE-PROOFPROGRAMME PROFILE 4.0?

7.      DISCUSSION:

-         taking into account the methodological issues, the authors should reconsider the manuscript section by section including limitations according to my previous comments and then revise accordingly! 

-         A subsection before “Future implications” before the sub-section of “limitations” would be shown!

8.      CONCLUSIONS: After reconsideration, this study will provide more important and sufficient conclusions!

-          

9.      REFERENCES: improved [16 & 18]

Reviewer 3 Report

Comments and Suggestions for Authors

Dear Authors ,

The study you carried out includes an important topic for nursing and its application based on specific roles that allow us to be measured in order to produce conclusions applicable to daily nursing practice.

However, it has some problems.

1 your introduction is well written and relevant.

2 the methodology you followed has been adequately described but the use of social media creates problems in the validity of the cross-section. I suggest you add a paragraph after the discussion and mention the problems of your study such as the one I mention, the small sample and anything else you think would help the reader.

3 the results are not written clearly especially in 3.5 where the correlations are not clear I suggest that you rewrite this paragraph, in general I would suggest that you look again at your results and render them with greater clarity as well as render tables 3 and 5 better

4 the discussion, regarding the comparison of other studies with yours is quite limited; I understand that the subject has a limited literature but this should also be mentioned in the limitations of your study. Especially from line 345 to the end of the discussion I think it should not be included in this chapter. I suggest you revise it and improve it to make it more understandable for readers.

5 the conclusions are very well written.

Round 2

Reviewer 1 Report

Comments and Suggestions for Authors

Introduction

The authors made corrections in the introduction. However, they do not specify which factors influence job satisfaction - even if it were the satisfaction of nurses in hospitals. I recommend to supplement.

Methods

2.2 Questionnaire

p.142-152 The description of methods for assessing job satisfaction is better than in the first manuscript. I recommend adding the variable "satisfaction with work packet, as the authors state in the cover letter - ("Satisfaction with work packet stands for the satisfaction with the content of all nurses' performances throughout the day, and relates to roles, tasks and responsibilities") .

Table 2a – I do not agree with the opinion of the authors. Extent to time spent to roles and contribution to satisfaction are different statistical variables - data on Cronbach alpha must be reported separately, not as a whole. Intra-rater reliability was implemented in isolation, therefore the input Cronbach alpha must also be evaluated separately for these variables.

Discussion

The authors of the manuscript state: "Moreover, scatter plots of the analyzes revealed two clear outliers that had a negative influence on the results." In other words, outliers significantly influence the presented results, therefore they should have been eliminated before the analysis, and the results should have been presented without outliers.
